# VILLAGE-NET CLUSTERING: A NOVEL UNSUPERVISED CLUSTERING METHOD

## ABSTRACT

We present "Village-Net Clustering," a novel unsupervised clustering algorithm designed for effectively clustering complex manifold data. The algorithm operates in two primary phases: first, utilizing K-Means clustering, it divides the dataset into distinct "villages." Subsequently, a weighted network is created, where each node represents a village, capturing their proximity relationships. To attain the optimal clustering, we cluster this network using the Walk-likelihood Community Finder (WLCF), a community detection algorithm developed by one of our team members. An important feature of Village-Net Clustering is its ability to autonomously determine the optimal number of cluster. Extensive benchmarking on real datasets with known ground-truth labels showcases its competitive performance, particularly in terms of the normalized mutual information (NMI) score, when compared to state-of-the-art methods. Additionally, the algorithm demonstrates impressive computational efficiency, boasting a time complexity of $O(N * k * d)$, where $N$ signifies the number of instances, $k$ represents the number of villages and $d$ represents the dimension of the dataset, making it well-suited for effectively handling large-scale datasets.

## 1 INTRODUCTION

Unsupervised clustering techniques enable the study of the inherent patterns, structures, and relationships within datasets. Unlike supervised methods, unsupervised approaches require little to no prior information about the dataset. This makes them particularly useful in information-limited settings where data is unlabeled or labels are not wholly trustworthy. Understanding the structure of data in the absence of additional knowledge is a fundamental problem in machine learning, with extensive applications in domains ranging from precision medicine (Cowley et al. (2023)) to cybersecurity Bierbrauer et al. (2021). At its core, the unsupervised clustering task is a problem of defining some notion of similarity onto a dataset and further, defining reasonable partitions on the basis of that similarity.

Many methods have been developed to tackle this problem. Traditional methods such as K-Means clustering (MacQueen (1967) , Lloyd (1982)), fuzzy C-Means clustering (Bezdek et al. (1984)) and K-Medoid (Ng & Han (2002)) clustering assume that a cluster is centred around a point. The data is clustered by minimizing the squared distance of the datapoints from their respective centers. Although these methods are simple and computational efficient, their inherent assumption of linear separability may not hold for real-life datasets that often exhibit complex and non-linear structures (Singh et al. (2011)).

In response to the limitations of conventional clustering techniques, several advanced methods have emerged to address the challenges posed by complex datasets. Two notable approaches are DBScan (Ester et al. (1996)) and OPTICS (Ankerst et al. (1999)), which are specifically designed to cluster data based on spatial density. While these methods excel in handling datasets with non-convex structures, the main disadvantage is that they are extremely sensitive to noise and the choice of hyperparameters (Dang (2015)).

Another avenue of advancement involves utilizing kernels to transform data into a high-dimensional feature space (Filippone et al. (2008)). This transformation empowers traditional algorithms such as K-Means (Dhillon et al. (2004)) and fuzzy C-Means (Ding & Fu (2016)) to achieve a linear

partitioning in this enriched feature space, enabling them to capture intricate patterns within the data.

For datasets with complex manifold structures, specialized techniques like mumCluster (Wang et al. (2010)) and K-Manifold (Souvenir & Pless (2005)) have been developed. These methods are tailored to efficiently cluster data that exhibits intricate geometric configurations. Additionally, there are dimensionality reduction techniques like t-SNE (van der Maaten & Hinton (2008)), UMAP (McInnes et al. (2018)), Spectral Clustering Ng et al. (2001), and Diffusion Maps (Coifman & Lafon (2006)). These methods provide lower-dimensional embeddings of datasets, making it possible to apply traditional clustering algorithms to multiple clusters.

However, a significant drawback of many advanced methods is their computational intensity. They often require the construction of large similarity or dissimilarity matrices for each data point or involve the identification of nearest neighbors, which can be computationally demanding.

In this paper, we aim to address the challenge of clustering datasets with non-convex cluster structures while maintaining computational efficiency and scalability. To achieve this, we introduce "VillageNet clustering," inspired by our previous work, MapperPlus (Datta et al. (2023)), which combines the topological data analysis (TDA) algorithm Mapper (Singh et al. (2007)) with Walk-likelihood Community Finder (Ballal et al. (2022)), a community detection algorithm developed by one of the authors. VillageNet clustering stands out by constructing a coarse-grained graph where each node, referred to as a "village," is formed through K-Means clustering. This approach strikes a balance between speed and accuracy, harnessing the power of K-Means clustering to identify meaningful partial-clusters within the dataset that capture the structure of the dataset while avoiding the complexities of extensive hyperparameter tuning.

An essential advantage of our method is its algorithmic efficiency, with a complexity of $O(N*k*d)$, where $N$ represents the number of datapoints in the dataset, $k$ stands for the number of villages created during the clustering process and $d$ denotes the dimensionality of the data.

This computational complexity profile ensures that our approach is scalable, making it suitable for handling datasets of varying sizes and dimensions. We have successfully applied VillageNet clustering to real-world datasets and have observed competitive performance when compared to state-of-the-art methods.

## 2 METHOD

### 2.1 PROBLEM FORMALISM

Consider a dataset $X$ where each point $\overrightarrow{x}_i, i = 1, 2, ..., N$ belongs to a $d$-dimensional Eucledian space $\mathcal{X}$ that has been divided into $m$ clusters $C_1, C_2, ..., C_m$ such that the cluster boundaries are continuous but not necessarily linear. Furthermore, within each cluster, the distribution of data points is such that the density is higher in the core of the cluster and lower on the cluster's boundary resulting in natural boundaries within the data space. Assuming that we have no information of the cluster identity of each instance or the number of cluster, our aim is as follows

1. Determine the number of clusters that best represent the underlying structure of the data.

2. Categorize each data point $i$ into one of the $m$ clusters based on its proximity and similarity to other data points within the same cluster.

### 2.2 STRATEGY

Our approach to solve the above stated problem is mainly built upon the following fundamental assumption:

*There exists a value of $k$ smaller than $N$ such that employing the K-means clustering with $k$ clusters can shatter the instrinsic clusters of dataset $X$ by $k$ Voronoi cells.*

This implies that given for each partial-cluster $V$ obtained from the K-means clustering, there exists an intrinsic cluster $C$ such that

$$\frac{\#\{i \in V : \mathcal{L}_i = C\}}{|V|} > 1 - \epsilon \tag{1}$$

for $\epsilon \ll 1$, where $\mathcal{L}_i$ refers to the cluster identity of the datapoint $i$ corresponding to the ground truth. Note that for this condition to hold true perfectly, that is for $\epsilon = 0$, $k$ should be greater than or equal to the anticipated Vapnik–Chervonenkis (VC) dimension of the dataset $X$ (Blumer et al. (1989), Vapnik (2000)).

Expanding on this idea, if we cluster the dataset into $k$ partial-clusters in a manner that each partial-cluster successfully shatters the dataset into meaningful subgroups, our task becomes simpler. This reduces the complexity of the problem to accurately clustering this set of partial-clusters. We term these partial-clusters as "villages." This initial step of our algorithm focuses on creating these village-level clusters, which act as a coarse-grained representation of the dataset.

To cluster this set of villages effectively, we construct a weighted graph, where each node represents a village where the edge weights, denoted as $A_{UV}$, encode the proximity relationships between the data points in villages $U$ and $V$. To cluster this graph, we utilze the Walk Likelihood Community Finding (WLCF) algorithm, which was developed by one of the authors. Finally, since we assume that each village mostly contains data points from a single category each data point is assigned the cluster identity corresponding to the cluster identity of the village it belongs to.

## 2.3 WALK-LIKELIHOOD COMMUNITY FINDER

The Walk-Likelihood Community Finder (WLCF) is an innovative community detection algorithm introduced by Ballal, Kion-Crosby, and Morozov \cite{WLCF}. Leveraging the inherent properties of random walks in graphs, WLCF excels in automatically determining the optimal number of communities within a graph, eliminating the need for prior knowledge about the community count. Notably, WLCF has demonstrated superior or comparable performance to state-of-the-art community-detection algorithms such as Louvain's algorithm and Non-negative Matrix Factorization (NMF).

At its core, WLCF employs the Walk-Likelihood Algorithm (WLA), which iteratively refines the community structure of the graph. The algorithm is grounded in a fundamental assumption about a discrete-time random walk on a network with weighted edges, denoted as \$w\_ij}\$, where $w_{ij}$ represents the rate of transmission from node $i$ to $j$. The random walker's movement is probabilistic, with the probability of jumping to its nearest neighbor $j$ given by $P(i \rightarrow j) = w_{ij}/w_i$, where $w_i = \sum_{k \in nn(i)} w_{ik}$ denotes the connectivity of node $i$, and the sum is over all nearest neighbors of node $i$.

Assuming the presence of a community $C_j$, WLA incorporates the notion of a Poisson distribution to model the probability of a random walker making $\mathcal{K}$ returns to node $i$ within the community $C_j$. The distribution is expressed as:

$$P(\mathcal{K}|\ell_j) = \mathcal{P}\left(\mathcal{K}, \frac{w_n \ell_j}{W_j}\right) = \frac{1}{\mathcal{K}!} \left(\frac{w_n}{W_j}\ell_c\right)^{\mathcal{K}} e^{-(w_n/W_j)\ell_j} \tag{2}$$

where $W_j = \sum_{i}^{i \in C_j} w_i$ is the weighted size of all nodes in community $C_j$. Leveraging Bayesian formalism, this probability is utilized to derive the posterior probability $P(i \in C_j)$ for each node $i$ and each community $C_j$. This information is iteratively employed to update the assignment of nodes to communities until a stable community configuration is reached.

WLCF employs global moves, involving bifurcation and merging of communities, to predict the optimal number of communities. The Walk-Likelihood Algorithm (WLA) is then utilized to refine node-community assignments at each step of the process.

## 2.4 ALGORITHM

The algorithm can be described in the following steps as shown in Fig. 1

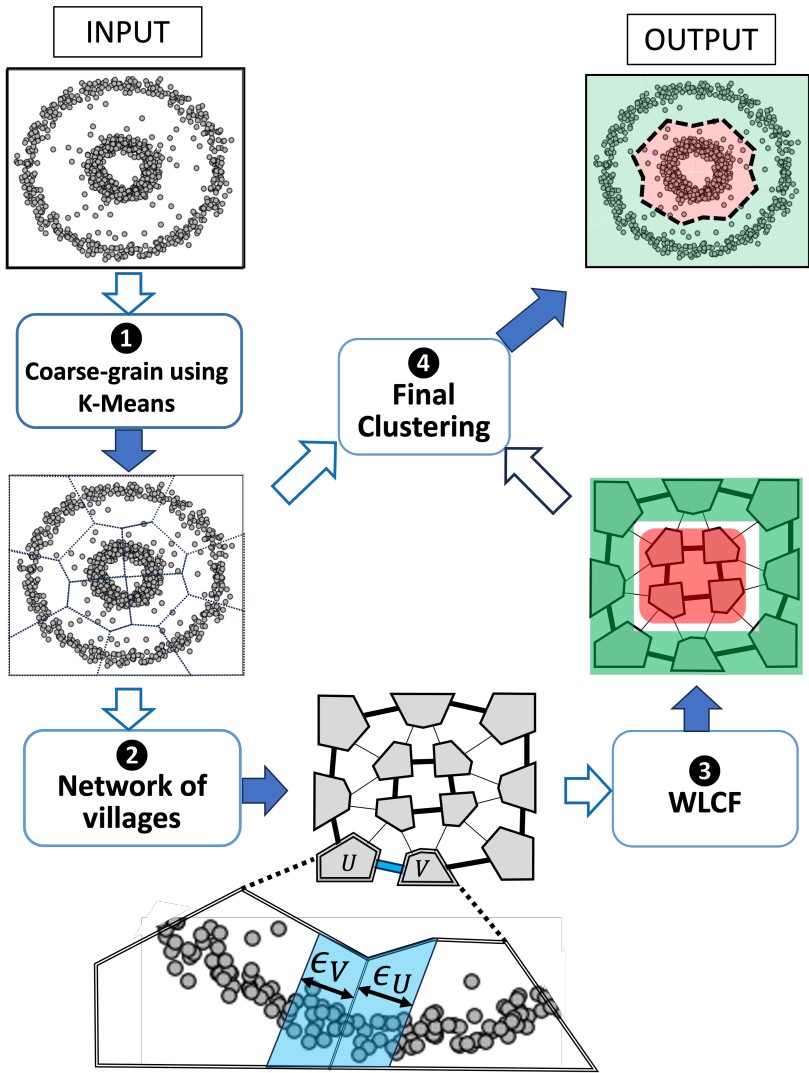

Figure 1: Sequential Stages of Village-Net Clustering

1. **K-Means clustering:** The initial raw dataset undergoes division into $k$ separate partial-clusters employing the K-Means clustering technique. The resultant partial clusters are referred to as "villages". Thus, each instance $i$ is assigned a unique village $V_i$ such that $i \in V_i$.

2. **Network of Villages:** For each datapoint $i$ and a village $U$ such that $i \notin U$, we define

$$\mathcal{D}(i, U) = \min\left\{|\overrightarrow{y}| : |\overrightarrow{X}_i - \overrightarrow{y} - \overrightarrow{\mu}_U| = |\overrightarrow{X}_i - \overrightarrow{y} - \overrightarrow{\mu}_{V_i}|\right\} \qquad (3)$$

where $\overrightarrow{\mu}_U$ and $\overrightarrow{\mu}_{V_i}$ are the centroids of the villages $U$ and $V_i$ respectively. Thus, $\mathcal{D}(i, U)$ is distance by which the point $\overrightarrow{X}_i$ needs to be shifted such that it is equidistant from $\overrightarrow{\mu}_U$ and $\overrightarrow{\mu}_{V_i}$. The direction of the shift $\overrightarrow{y}$ for $i$ would be along the vector joining the $\overrightarrow{\mu}_{V_i}$ and $\overrightarrow{\mu}_U$. Thus, $\mathcal{D}(i, U)$ is the projection

$$\mathcal{D}(i, U) = \left(\overrightarrow{X}_i - \frac{1}{2}(\overrightarrow{\mu}_U + \overrightarrow{\mu}_{V_i})\right) \cdot \left(\frac{\overrightarrow{\mu}_{V_i} - \overrightarrow{\mu}_U}{|\overrightarrow{\mu}_{V_i} - \overrightarrow{\mu}_U|}\right). \qquad (4)$$

Thus, $\mathcal{D}(i, U)$ provides a distance for each datapoint $i$ from the village $U$, based on which we define the exterior of the village $U$ as

$$U^{(E)} = \{i | \mathcal{D}(i, U) < \epsilon_U \text{ and } i \notin U\}. \qquad (5)$$

for a given hyperparameter $\epsilon_u$. We construct a weighted network where each node represents a village. The edge weight between villages $U$ and $V$ is given by

$$A_{UV} = |U^{(E)} \cap V| + |U \cap V^{(E)}| \tag{6}$$

Geometrically, $A_{UV}$ symbolizes the count of points situated within the region formed in the region of village $U$ and village $V$ (as illustrated in Fig 1) delineated by two parallel $(d-1)$-dimensional hyperplanes, each parallel to the decision boundary. Note that $A$ represents the adjacency matrix of the network.

For each village $U$ we choose the hyperparameter $\epsilon_U$ such that $|U^{(E)}| = r$. This condition establishes consistency in the neighborhood of each village. Here, $r$, often set between 5 and 100, bears resemblance to the perplexity parameter in t-SNE. This condition is satisfied by partially sorting the list $[\mathcal{D}(i, U) \; \forall i \notin U]$ to obtain the indices of the minimum $r$ values.

3. **WLCF:** The network of villages is subjected to the Walk-Likelihood Community Finder, which partitions the network of villages into disjoint communities of villages.

4. **Final clustering:** To achieve the ultimate dataset clustering, the datapoints affiliated with villages within each community are amalgamated. Given the non-overlapping nature of the villages, the final clustering is inherently disjoint.

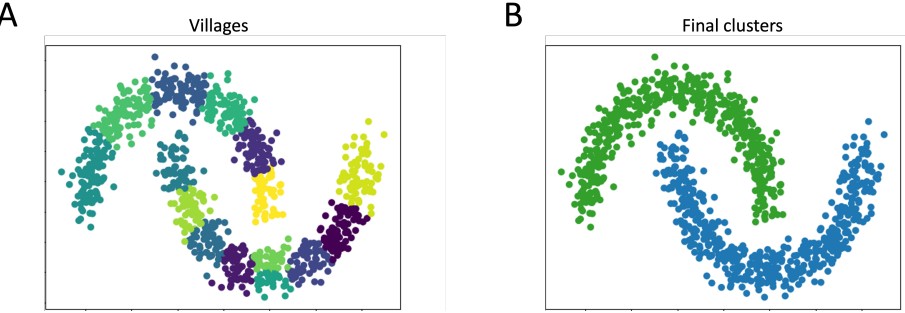

Figure 2: Visualization of Village-Net Clustering on Two-Moons dataset

The Village-Net Clustering is illustrated using the two moons dataset with parameters set at $k = 15$ and $r = 20$. Fig 2A displays the 15 villages obtained using K-Means clustering (Step 1), while Steps 2-4 show the partitioned villages resulting in the final clustering (Fig 2B).

## 2.5 Algorithm Complexity

The time complexity of each step in our algorithm is as follows:

1. **K-Means clustering:** This step is the most time-consuming in the algorithm. While K-Means is known to be an NP-hard problem, our approach focuses on achieving a locally optimal solution rather than an exact one, significantly reducing computational demands. The time complexity of this step is approximately $O(N * k * d)$, where $N$ represents the number of instances in the dataset, $k$ is the number of villages, and $d$ is the dimensionality of the dataset.

2. **Network of villages:** The most computationally expensive task in this step is the partial sorting of lists, which is performed using the argpartition function in numpy. The algorithmic complexity of partial sorting is $O(N)$, and this operation is carried out $k$ times, once for each village. Thus, the overall time complexity of this step is $O(N * k)$.

3. **WLCF:** Empirical observations suggest that the time complexity of the walk-likelihood community finder is approximately $O(k^{1.5})$, where $k$ represents the number of nodes in the network.

4. **Final clustering:** This step is the least computationally intensive and only requires $O(N)$ time.

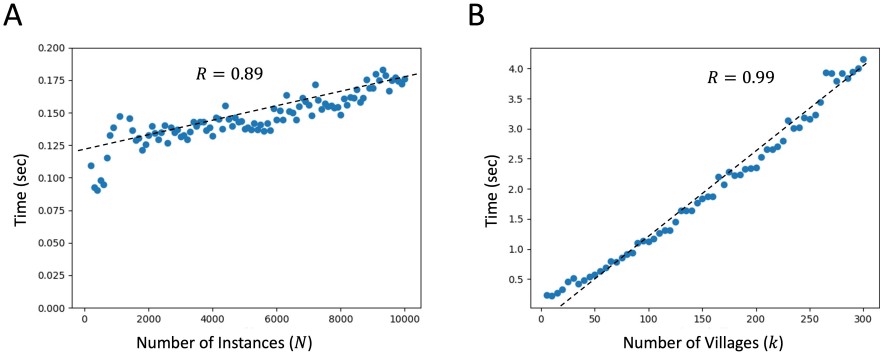

Figure 3: Wall Time Analysis of Village-Net Clustering on Various Implementations of the Two-Moons Dataset

Thus, in the limit that $k \ll N$, he overall time complexity of the algorithm approaches $O(N * k * d)$. To validate this, Fig 3 displays the wall time of Village-Net on different implementations of the two moons dataset. Fig 3A illustrates variations in the dataset size with a constant $k = 50$, while Fig 3B demonstrates varying numbers of communities for a fixed dataset size. Empirically, the observed time complexity aligns with the theoretically predicted one, as depicted in Fig 3.

## 2.6 EFFECT OF HYPERPARAMETERS

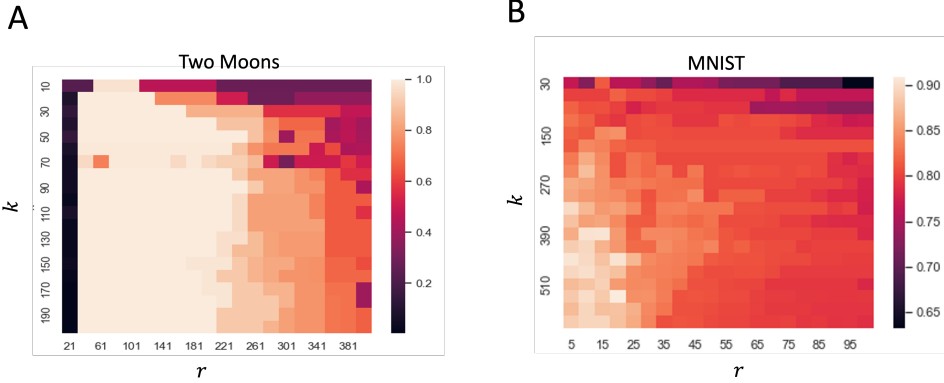

Figure 4: Comparison of clusters obtained by Village-Net Clustering on different hyperparameters with the ground truth on Digits dataset and Two-Moons dataset

Fig 4 provides insights into the impact of Village-Net hyperparameters on the final clustering results, evaluated on both the artificial two moons dataset and the real-world digits dataset. The assessment is based on the comparison of obtained clusters with the ground truth, measured by Normalized Mutual Information (NMI). The algorithm involves two crucial hyperparameters:

1. **$k$**: The performance of our algorithm is closely tied to the choice of parameter $k$. Since, in step 4 of our algorithm, each data point within a village is assigned the same cluster identity as the village itself the accuracy of our algorithm in clustering the dataset is contingent on how effectively K-Means clustering can shatter the true labels of the dataset. This accuracy tends to improve with larger values of $k$ as observed in Fig4. However, as $k$ increases, so does the algorithm's runtime linearly.

   Therefore, selecting the optimal value for $k$ involves a trade-off. On one hand, a higher $k$ improves clustering accuracy; on the other hand, it escalates computational time. Thus,

the ideal choice for $k$ lies in finding the minimum number of K-Means clusters that can approximately represent the underlying structure of the data.

2. **$r$:** This hyperparameter $r$ plays a critical role in shaping the connectivity within the village graph. In the extreme scenario where $k$ equals $N$, each individual data point becomes a village, essentially transforming the network of villages into an $r$-nearest neighbor graph of the dataset. Conversely, when $k < N$, the village network provides a coarser representation of this $r$-nearest neighbor graph.

   The value of $r$ is pivotal in expressing the connectivity between villages. It needs to be chosen judiciously, considering the balance between granularity and coarseness in representing local structures. If $r$ is too large, it can lead to a dilution of information about the local structures within villages, as observed in Fig 4. This dilution poses a challenge in obtaining correct clusters, emphasizing the importance of appropriately tuning $r$ to capture the desired level of connectivity without sacrificing the fidelity of local information within villages.

## 2.7 SOFTWARE AVAILIBILITY

The software is publicly available at https://github.com/lordareicgnon/VillageNet.

## 3 EXPERIMENTS

### 3.1 DATASETS

We evaluate the performance of Village-Net Clustering on both eight real-world datasets (Table 1). Each dataset had known labeled clusters.

|  | Number of observations | Number of attributes | Known number of clusters |
|---|---|---|---|
| Pen Digits | 1797 | 64 | 10 |
| MNIST | 70000 | 784 | 10 |
| FMNIST | 70000 | 784 | 10 |
| Letters | 20000 | 16 | 26 |
| Dry Beans | 13611 | 16 | 6 |
| Wine | 178 | 13 | 3 |
| Wifi | 2000 | 7 | 4 |
| Breast Cancer | 569 | 30 | 2 |

Table 1: Summary of the publicly-available real-world datasets used for method comparison

### 3.2 COMPETING METHODS

We conducted a performance comparison between Village-Net Clustering and several other clustering algorithms, namely KMeans, DBScan and UMAP-enhanced DBScan, Birch, OPTICS and Agglomerative Clustering. UMAP-enhanced clustering, in particular, leverages UMAP as a preprocessing step on the dataset. Subsequently, DBScan is applied to the UMAP-generated embedding for clustering purposes. These algorithm selections were made based on their widespread use and their ability to efficiently cluster large datasets within reasonable time frames. We applied Village-Net Clustering with varying village numbers ($k = 50, 100, 200,$ and $500$) but fixed $r = 20$ on large datasets. On small datasets we used $r = 20$ and $k = 20, 50, 100, 200$. Note that K-Means and Agglomerative Clustering require the number of clusters to be given as an input. BIRCH, OPTICS and Agglomerative clustering ran for more than 2 hours on MNIST and FMNIST without any results.

### 3.3 RESULTS

In our clustering algorithm evaluation, we utilized the Normalized Mutual Information (NMI) metric and Adjusted Rand Index (ARI), both of which range from 0 to 1, with 1 representing optimal performance and 0 indicating the poorest. For Village-Net Clustering we observed a consistent

| Methods/ Datasets | | MNIST (m*=10) | | | FMNIST (m*=10) | | | Letters (m*=26) | | | Dry Beans (m*=6) | | |
|---|---|---|---|---|---|---|---|---|---|---|---|---|---|
| | | NMI | ARI | m | NMI | ARI | m | NMI | ARI | m | NMI | ARI | m |
| Village-Net | k=50 | 0.65 | 0.43 | 5 | 0.59 | 0.51 | 6 | 0.51 | 0.19 | 12 | 0.63 | 0.56 | 5 |
| | k=100 | 0.71 | 0.58 | 8 | **0.65** | **0.59** | 9 | 0.54 | 0.22 | 13 | 0.65 | 0.6 | 7 |
| | k=200 | 0.79 | 0.71 | 9 | 0.64 | 0.58 | 9 | **0.56** | **0.25** | 19 | 0.64 | 0.58 | 8 |
| | k=500 | 0.82 | 0.75 | 9 | 0.64 | 0.58 | 9 | 0.55 | 0.22 | 13 | 0.64 | 0.58 | 10 |
| K-Means | | 0.51 | 0.34 | | 0.51 | 0.45 | | 0.36 | 0.14 | | **0.70** | **0.65** | |
| DBScan | | 0.46 | 0.29 | 6 | 0.32 | 0.3 | 4 | 0.39 | 0.12 | 6 | 0.51 | 0.43 | 4 |
| UMAP +DBScan | | **0.88** | **0.82** | 8 | 0.59 | 0.49 | 5 | 0.49 | 0.15 | 5 | 0.57 | 0.52 | 3 |
| BIRCH | | | | | | | | 0.17 | 0.04 | 3 | 0.53 | 0.32 | 3 |
| OPTICS | | | | | | | | 0.23 | 0.00 | 575 | 0.12 | 0.0 | 155 |
| Agglomerative | | | | | | | | 0.43 | 0.16 | | 0.69 | 0.6 | |

Table 2: Comparative Analysis of Unsupervised Clustering Algorithms on Large Datasets

| Methods/ Datasets | | Digits (m*=10) | | | Wine (m*=3) | | | Breast Cancer (m*=2) | | | Wifi (m*=4) | | |
|---|---|---|---|---|---|---|---|---|---|---|---|---|---|
| | | NMI | ARI | m | NMI | ARI | m | NMI | ARI | m | NMI | ARI | m |
| Village-Net | k=20 | 0.81 | 0.61 | 8 | 0.7 | 0.69 | 3 | **0.64** | **0.75** | 2 | 0.88 | 0.89 | 4 |
| | k=50 | 0.83 | 0.63 | 8 | 0.74 | 0.75 | 3 | 0.38 | 0.27 | 4 | 0.84 | 0.85 | 5 |
| | k=100 | 0.85 | 0.69 | 9 | 0.84 | 0.85 | 3 | 0.4 | 0.34 | 4 | 0.84 | 0.82 | 5 |
| | k=200 | 0.89 | 0.73 | 10 | | | | 0.37 | 0.24 | 6 | 0.83 | 0.82 | 5 |
| K-Means | | 0.74 | 0.66 | | **0.88** | **0.9** | | 0.55 | 0.67 | | 0.8 | 0.82 | |
| DBScan | | 0.71 | 0.61 | 6 | 0.7 | 0.72 | | 0.40 | 0.43 | | 0.74 | 0.77 | |
| UMAP + DBScan | | **0.91** | **0.8** | 9 | 0.71 | 0.73 | | 0.43 | 0.51 | | 0.82 | 0.85 | |
| BIRCH | | 0.54 | 0.3 | 3 | 0.79 | 0.79 | 3 | 0.44 | 0.53 | 3 | 0.71 | 0.57 | 3 |
| OPTICS | | 0.37 | 0.06 | 37 | 0.19 | 0.04 | 4 | 0.03 | 0.05 | 2 | 0.19 | 0.0 | 48 |
| Agglomerative | | 0.87 | 0.79 | | 0.79 | 0.79 | | 0.46 | 0.58 | | **0.9** | **0.91** | |

Table 3: Comparative Analysis of Unsupervised Clustering Algorithms on Small Datasets

trend: as the number of villages increased, NMI as well as ARI scores also increase, eventually stabilizing at a constant value.

Table 2 and 3 presents a summary of algorithm performance results. Village-Net Clustering consistently demonstrated impressive performance across all datasets. Notably, it outperformed other algorithms on the FMNIST and Letters and is the second best in all the other datasets. In the case of the Dry Beans dataset, K-Means showed a slight advantage over Village-Net, while for the Digits dataset, DBScan on UMAP space marginally outperformed Village-Net. Notably, only on the MNIST dataset did DBScan on UMAP notably surpass Village-Net. Village-Net's prediction of the number of clusters is not too far from the real value.

Table 4 provides a comprehensive overview of algorithm runtimes, revealing a striking efficiency in Village-Net Clustering across all datasets. Note that since BIRCH, OPTICS and Agglomerative clustering diverge on MNIST and FMNIST, they are not included in Table 4. Notably, Village-Net consistently outperforms other algorithms by a substantial margin, showcasing significantly shorter runtimes. Remarkably, in certain instances, Village-Net even surpasses the speed of K-Means, traditionally recognized for its efficiency.

This exceptional speed can be attributed to the unique K-Means implementation within Village-Net. The algorithm prioritizes local solutions, opting for efficiency over the pursuit of a globally optimal initialization. As a result, Village-Net bypasses the computationally intensive K-Means++ initialization step, a hallmark of the standard K-Means algorithm. The noteworthy efficiency of the algorithm is a key aspect of the novelty inherent in Village-Net Clustering. It excels in rapidly clustering large and intricate datasets, offering a significant advantage in terms of time efficiency.

| Methods/Datasets | | Pen Digits | MNIST | Letters | Dry-Beans | FMNIST |
|---|---|---|---|---|---|---|
| Village-Net | k=50 | <1 | 4 | <1 | <1 | 5 |
| | k=100 | <1 | 7 | <1 | <1 | 8 |
| | k=200 | <1 | 10 | <1 | <1 | 11 |
| | k=500 | 2 | 24 | 2 | 2 | 25 |
| K-Means | | <1 | 25 | 5 | 10 | 22 |
| DBScan | | <1 | 247 | 6 | 14 | 300 |
| UMAP + DBScan | | 11 | 64 | 40 | 35 | 58 |

Table 4: Comparing Unsupervised Clustering Algorithms in Terms of Computational Runtime (Seconds)

In summary, across all datasets, Village-Net Clustering either produced the best results or competitive results with the added advantage of significantly reduced computational time.

## 4 CONCLUSION AND FUTURE WORK

Village-Net Clustering has demonstrated excellent performance on real-world datasets, excelling in terms of both similarity to the ground truth and runtime efficiency. It strikes a favorable balance between accuracy and speed. However, it does have a significant limitation: its accuracy is constrained by the expressive capacity of the K-Means algorithm to shatter the intrinsic clusters of a dataset effectively.

Looking ahead, our future plans involve developing an iterative approach. This approach aims to enhance the clustering process by gradually increasing the ability of the "villages" (clusters) to effectively divide and capture the complexities of the dataset, potentially improving overall clustering accuracy.

### ACKNOWLEDGMENTS

This work has been supported by the grants from United States National Institutes of Health (NIH) R01HL149431, R01HL90880, R01HL123526, R01HL141460, R01HL159993, and R35HL166575. The authors gratefully acknowledge Gegory Depaul for his expertise and help. Without him, this project was impossible.

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
