# OpenReview forum: "Village-Net clustering: A novel unsupervised manifold clustering method"
_ICLR.cc/2024/Conference — Submitted to ICLR 2024_

### Official Review · Reviewer_Uwn5 · 2023-10-27

**Soundness:** 2 fair
**Presentation:** 2 fair
**Contribution:** 2 fair
**Rating:** 3
**Confidence:** 5

**Summary:**

The paper presents a clustering method that is based on overclustering (using k-means) followed by aggregation of the inital subclusters (called villages). Subcluster aggregation is performed by first computing a similarity measure between pairs of subclusters that measures how the extension of  the first cluster creates overlap with the second cluster. The similarity matrix is then employed by the Walk Likelihood Community Finder algorithm (WLCF) that essentially provides the final clusters.

**Strengths:**

The definition of the similarity measure A_{UV} between subclusters U and V seems novel and interesting, however it includes a hyperparameter.

**Weaknesses:**

- The method depends on hyperparameters: initial number of clusters k and and parameter r that establishes the neighborhood of each subcluster.
- It is not clear whether the method estimates the final number of clusters or the number of clusters is given as input to the method.
- Experimental part is weak and could be improved (see questions below).

**Questions:**

1) If the method estimates the number of clusters, then results should be provided on how close are the estimations to the actual values.
2) Since the similarity matrix is computed, typical agglomerative approaches (eg. single linkage) or spectral clustering methods could have been used instead of the WLCF algorithm. What is the advantage of WLCF?
3) Several approaches based on overclustering and cluster aggregation have been proposed. Some of them could have been used in the experimental study.
4) A short presentation of the WLCF method should be included. Does this method contain hyperparameters? What is the theoretical complexity (not the empirical complexity) of the method?
5) The number of attributes in Pendigits dataset is not 64 but 16.
6) Experiments using additional datasets could have been be included to draw more reliable conclusions on the performance of the method.

---

> ### Author Response · Authors · 2023-11-23
>
> Dear Reviewer Uwn5,
> Thank you for your feedback.
> Weaknesses:
> 1.     The method depends on hyperparameters: initial number of clusters k and and parameter r that establishes the neighborhood of each subcluster.
>
> We have added a detailed analysis on the effects of the parameter r. Additionally, While it's acknowledged that the VillageNet clustering method depends on the initial performance of K-Means clustering, we emphasize that our primary goal is not to cluster the data effectively into k villages. Instead, we use K-Means to form a network of villages, leveraging the information from this clustering to achieve the final results. Specifically, choosing k such that k≫m ensures meaningful clustering.
>
> 2.      It is not clear whether the method estimates the final number of clusters or the number of clusters is given as input to the method.
> Our method estimates the number of clusters.
>
> 3.     Experimental part is weak and could be improved (see questions below).
>
> We address this below.
>
> Questions:
> 1.       If the method estimates the number of clusters, then results should be provided on how close are the estimations to the actual values.
>
> We agree and have demonstrated this using a number of metrics, specifically NMI, adjusted RAND, among others.
>
> 2.    Since the similarity matrix is computed, typical agglomerative approaches (eg. single linkage) or spectral clustering methods could have been used instead of the WLCF algorithm. What is the advantage of WLCF?
> WLCF is shown to be better than state of the art algorithms like Louvain's algorithm. Plus unlike spectral clustering, it does not need the number of communities as an input, instead the number of communities is given as an output.
>
> 3.     Several approaches based on overclustering and cluster aggregation have been proposed. Some of them could have been used in the experimental study.
>
> We agree and have included a comparison to the BIRCH algorithm, which does overclustering and cluster aggregation.
>
> 4.      A  short presentation of the WLCF method should be included. Does this method contain hyperparameters? What is the theoretical complexity (not the empirical complexity) of the method?
> In the revised version, a more detailed explanation of WLCF is included. The theoretical complexity of WLCF has not been calculated in the paper.
>
> 5.     The number of attributes in Pendigits dataset is not 64 but 16.
> We used the sklearn version of digits. It has 64 (8X8) parameters
>
> 6.     Experiments using additional datasets could have been be included to draw more reliable conclusions on the performance of the method.
> The revised version contains this.

---

### Official Review · Reviewer_coeZ · 2023-10-27

**Soundness:** 1 poor
**Presentation:** 2 fair
**Contribution:** 2 fair
**Rating:** 5
**Confidence:** 4

**Summary:**

The main contribution of this paper is that it proposed a clustering algorithm named Village-Net Clustering, which aims to address the challenge of clustering datasets with non-convex cluster structures while maintaining computational efficiency and scalability. The algorithm is executed in four steps: K-Means initialization, construction of a weighted villages network, partitioning the network of villages into disjoint communities, and final clustering.

**Strengths:**

1. This paper is well-written and easy to follow.
2. The proposed algorithm is technically sound.
3. The algorithm flow in this paper (Figure 1) is vivid and clear.

**Weaknesses:**

1. The novelty of this work is relatively limited, and it seems that only the construction of a village network is proposed in this paper.
2. The comparison algorithms used in the experimental part are few, and all of them are early clustering algorithms, so the comparison experiments with the clustering algorithms in recent years should be added.
3. According to the part of the effect of hyperparameters in this paper, the algorithm is affected by k and r, but there is no strategy to analyze the selection of hyperparameters, and r is fixed at 20 in the experiment part, and comparative experiments for hyperparameter analysis should be added.

**Questions:**

1. Compared with recent clustering algorithms, is the performance of the proposed algorithm still better?
2. The values of k and r have a great influence on the clustering performance. How to determine the values of hyperparameters for a new data set?
3. Can the convergence of the proposed algorithm be proved theoretically?
4. When evaluating the performance of the algorithm, Table 2 only takes NMI as an evaluation index. Can the author verify the effectiveness of the proposed algorithm under more evaluation indexes?
5. This paper uses the k-means algorithm for initialization. If the number of class clusters in the data set is large, such as greater than 100 or 1000, is this algorithm applicable?

---

> ### Author Response · Authors · 2023-11-23
>
> Dear Reviewer coeZ,
> Thank you for your valuable feedback.
>
> Weaknesses:
> 1.     The novelty of this work is relatively limited, and it seems that only the construction of a village network is proposed in this paper.
> We have expanded the discussion to include the broader contribution of our paper.
>
> 2.      The comparison algorithms used in the experimental part are few, and all of them are early clustering algorithms, so the comparison experiments with the clustering algorithms in recent years should be added.
>
> We have significantly expanded the list of comparison algorithms in the revised version of the paper.
>
> 3.       According to the part of the effect of hyperparameters in this paper, the algorithm is affected by k and r, but there is no strategy to analyze the selection of hyperparameters, and r is fixed at 20 in the experiment part, and comparative experiments for hyperparameter analysis should be added.
>
> We have included an extensive analysis of the effects of the hyperparameters.
>
> Questions:
> 1.      Compared with recent clustering algorithms, is the performance of the proposed algorithm still better?
> We have provided a better comparison of our method in the revised version. However because of lack of publicly available code, we found it hard to compare our algorithm to the recent one.
>
> 2.     Can the convergence of the proposed algorithm be proved theoretically?
> We haven’t found a method to do that yet. In the future, we plan to work on it.
>
> 3.     When evaluating the performance of the algorithm, Table 2 only takes NMI as an evaluation index. Can the author verify the effectiveness of the proposed algorithm under more evaluation indexes?
> 	We have expanded the list of evaluation metrics for making a more persuasive case.
>
> 4.        This paper uses the k-means algorithm for initialization. If the number of class clusters in the data set is large, such as greater than 100 or 1000, is this algorithm applicable?
> Our algorithm can work well as long as we have the condition m<<k<<N where m is the number of class clusters, k is the number of villages and N is the number of datapoints.

---

### Official Review · Reviewer_K5W5 · 2023-10-30

**Soundness:** 3 good
**Presentation:** 3 good
**Contribution:** 1 poor
**Rating:** 3
**Confidence:** 4

**Summary:**

In this paper, the authors proposed a clustering method. It first overclusters the data points into initial small clusters (village), then builds a graph on the villages, and finally applies an existing community detection methods to obtain the final clustering. Experiments show the effectiveness of the proposed method.

The authors violated the double blind rule. For instance, it is mentioned in Page 2 that the idea is inspired by their previous work, MapperPlus and WLCF is developed by one of the authors.

**Strengths:**

1. The proposed method seems to be accurate and efficient in practice.

**Weaknesses:**

1. The contribution of the paper is not so clear. The idea of over-clustering + merging is not new, which can be find in many previous methods, like BIRCH.
2. The proposed method is a combination of different techniques, but it is not clear why this combination is unique and which specific unsolved problem it can handle.
3. The comparison methods are weak, only K-means and DBSCAN.
4. The performance of the proposed method strongly depend on K-means and hyper-parameters.
5. The authors violated the double blind rule.

**Questions:**

See above weakness.

---

> ### Author Response · Authors · 2023-11-23
>
> Dear, Reviewer K5W5
> Thank you for your valuable feedback on our manuscript.
>
> 1.     The contribution of the paper is not so clear. The idea of over-clustering + merging is not new, which can be find in many previous methods, like BIRCH.
>
> We have expanded the discussion to include the broader contribution of our paper. We have also included a comparison to other methods, such as BIRCH.
>
> 2.       The proposed method is a combination of different techniques, but it is not clear why this combination is unique and which specific unsolved problem it can handle.
>
> We have explicitly provided a more detailed explanation of what makes Village-Net innovative, emphasizing its fast clustering capabilities for complex data.
>
> 3.      The comparison methods are weak, only K-means and DBSCAN.
>
> We have added more comparison methods that are unsupervised and computationally feasible.
>
> 4.     The performance of the proposed method strongly depend on K-means and hyper-parameters.
>
> 	While it's acknowledged that the VillageNet clustering method depends on the initial performance of K-Means clustering, we emphasize that our primary goal is not to cluster the data effectively into k villages. Instead, we use K-Means to form a network of villages, leveraging the information from this clustering to achieve the final results. Specifically, choosing k such that k≫m ensures meaningful clustering. We also include a detailed analysis of the key hyperparameters.

---

### Official Review · Reviewer_fUNF · 2023-10-30

**Soundness:** 2 fair
**Presentation:** 3 good
**Contribution:** 2 fair
**Rating:** 5
**Confidence:** 4

**Summary:**

In this work, the authors propose an unsupervised clustering method, Village-Net Clustering, whose core idea is to divide the dataset into multiple "villages" by K-Means algorithm, and then construct a weighted network among these "villages". Finally, the Walk-likelihood Community Finder (WLCF) algorithm is utilized to cluster the network, thus realizing the clustering of the original data.  The authors implemented experimental comparisons to demonstrate that the method achieves impressive computational efficiency on multiple datasets. The authors employ suitable methods and pose a clear research question.

**Strengths:**

The author describes the fundamental algorithm well; and they seem to give all relevant information to understand and reproduce their algorithm.

The paper outlines a novel clustering method which is capable of clustering complex manifold data.

Writing and presentation skill is well.

**Weaknesses:**

1.Where is the title of unsupervised manifold clustering reflected in the manuscript? The authors should have explicitly described what is innovative about Village-Net.
2.The authors should consider collecting more publicly available data to confirm the validity of their Village-Net Clustering.
3.The authors did not compare their method with latest state-of-the-art methods. They may need to compare Village-Net with other unsupervised manifold clustering methods.

**Questions:**

Major Concerns:
1.Where is the title of unsupervised manifold clustering reflected in the manuscript? The authors should have explicitly described what is innovative about Village-Net.
2.The authors should have discussed the effect on the model of the choice of the hyperparameter r. r is a key parameter in Village-Net, and the clustering effect is highly dependent on r. However, the manuscript does not provide any details on how to choose r.
3.The authors should consider collecting more publicly available data to confirm the validity of the Village-Net Clustering.
4.The authors mention Village-Net clustering outperformed other algorithms on the FMNIST and Letters and is the second best in all the other datasets. The manuscript only use NMI to evaluate the clustering, and also need to consider more evaluation metrics on the performance comparison is more persuasive.
5.The authors may need to compare Village-Net with other unsupervised manifold clustering methods.
6.How to improve the model generalization ability? For different datasets, the hyperparameter k needs to be adjusted individually to obtain superior results, indicating poor model generalization ability.

Minor Concerns:
1.How to divide a network of villages into disjoint village communities using WLCF? Please describe the WLCF algorithm in detail.
2.Please check that the formula symbols in the manuscript are correct. For example, Page 3 Formula (4).
3.Please check that punctuation is used correctly in the manuscript. For example, Page 1 line 1 ”Village-Net Clustering,” should be changed to “Village-Net Clustering”.
4.Page 4 line 9, “T-SNE” should be changed to “t-SNE”.
5.Page 5 line 12, “O(N ∗ k)” should be changed to “O(N ∗ k)”.

---

> ### Author Response · Authors · 2023-11-23
>
> Dear Reviewer fUNF,
> Thank you for your valuable feedback on our manuscript. We appreciate the time and effort you invested in providing detailed comments and suggestions. We have carefully considered each point raised and made the necessary revisions to enhance the quality and clarity of our work. Here is a coherent summary of our responses to your concerns:
>
>
> Weaknesses
> 1.     Where is the title of unsupervised manifold clustering reflected in the manuscript? The authors should have explicitly described what is innovative about Village-Net.
>
> We have explicitly provided a more detailed explanation of what makes Village-Net innovative, emphasizing its fast clustering capabilities for complex data.
>
>
> 2.      The authors should consider collecting more publicly available data to confirm the validity of their Village-Net Clustering.
>
> We acknowledge the importance of publicly-available data. As such, we have added a few public datasets and artificial datasets as benchmarks to our analysis.
>
> 3.       The authors did not compare their method with latest state-of-the-art methods. They may need to compare Village-Net with other unsupervised manifold clustering methods.
>
> We agree that we should have compared our methods with the latest state-of-the-art methods. We have run new experiments with other unsupervised clustering techniques in order to fully compare our method with others.
>
> Major Concerns:
> 1.        Where is the title of unsupervised manifold clustering reflected in the manuscript? The authors should have explicitly described what is innovative about Village-Net.
> We have explicitly provided a more detailed explanation of what makes Village-Net innovative, emphasizing its fast clustering capabilities for complex data.
>
> 2.         The authors should have discussed the effect on the model of the choice of the hyperparameter r. r is a key parameter in Village-Net, and the clustering effect is highly dependent on r. However, the manuscript does not provide any details on how to choose r.
>
> In the new version, we've included a detailed analysis of the hyperparameter "r" to provide a better understanding of its impact on the algorithm and clustering results.
>
>
> 3.         The authors should consider collecting more publicly available data to confirm the validity of the Village-Net Clustering.
>
> We acknowledge the importance of publicly-available data. As such, we have added a few public datasets and artificial datasets as benchmarks to our analysis.
>
> 4.         The authors mention Village-Net clustering outperformed other algorithms on the FMNIST and Letters and is the second best in all the other datasets. The manuscript only use NMI to evaluate the clustering, and also need to consider more evaluation metrics on the performance comparison is more persuasive.
>
> In response to your suggestion, we've expanded the set of evaluation metrics in the revised version to provide a more comprehensive assessment of the proposed algorithm's performance.
>
>
>  5.        The authors may need to compare Village-Net with other unsupervised manifold clustering methods.
>
> We agree that we should have compared our methods with the latest state-of-the-art methods. We have run new experiments with other unsupervised clustering techniques in order to fully compare our method with others.
>
>
> 6.          How to improve the model generalization ability? For different datasets, the hyperparameter k needs to be adjusted individually to obtain superior results, indicating poor model generalization ability.
>
> In general we expect stable results across a range of hyperparameter k.
>
> Minor Concerns:
> 1.      How to divide a network of villages into disjoint village communities using WLCF? Please describe the WLCF algorithm in detail.
>
> We've enhanced the clarity of the description of WLCF in response to your feedback.
>
>  2.       Please check that the formula symbols in the manuscript are correct. For example, Page 3 Formula (4). 3.Please check that punctuation is used correctly in the manuscript. For example, Page 1 line 1 ”Village-Net Clustering,” should be changed to “Village-Net Clustering”. 4.Page 4 line 9, “T-SNE” should be changed to “t-SNE”. 5.Page 5 line 12, “O(N ∗ k)” should be changed to “O(N ∗ k)”.
>
> We believe these revisions address the concerns raised, providing a more robust and comprehensive presentation of our work. We appreciate your insightful comments and hope the revised manuscript meets your expectations.

---

### Official Review · Reviewer_CGby · 2023-10-31

**Soundness:** 2 fair
**Presentation:** 2 fair
**Contribution:** 2 fair
**Rating:** 3
**Confidence:** 4

**Summary:**

This paper introduces an unsupervised clustering algorithm called "Village-Net Clustering" for effectively clustering complex manifold data. The algorithm consists of two main phases: first, it uses K-Means clustering to divide the dataset into distinct "villages", and then it creates a weighted network where each node represents a village, capturing their proximity relationships. To achieve optimal clustering, the network is clustered using the Walk-likelihood Community Finder (WLCF) algorithm. Experiments on real datasets show that the Village-Net Clustering algorithm possesses certain advantages in terms of clustering performance and computational efficiency.

**Strengths:**

1. This article introduces some new ideas. Firstly, it uses K-Means to construct coarse-grained "villages" for the raw dataset, and then redefines a distance calculation method between these villages.
2. High algorithm efficiency: VillageNet clustering has lower computational complexity, making it suitable for processing datasets of different scales and dimensions.
3. Ability to handle non-convex clustering structures: VillageNet clustering can handle datasets with non-convex clustering structures, capturing complex structures within the dataset.

**Weaknesses:**

1. Dependency on K-Means Clustering: The first step of the VillageNet clustering method involves using the K-Means clustering algorithm to create initial "villages." Therefore, the performance and results of K-Means clustering have a certain dependency.
2. Assumption about Data Distribution: VillageNet clustering assumes the presence of some local clustering structures within the dataset, and that these local structures can be successfully separated by the K-Means clustering algorithm. This assumption may not hold suit for most datasets, leading to inaccurate clustering results.
3. Lack of More Experimental Validation: The algorithm was only compared with a few traditional clustering algorithms, and the experimental results indicate that the proposed algorithm performs well only on two datasets. Therefore, it is difficult to demonstrate that the proposed algorithm exhibits competitive performance.
4. The evaluation metrics for the experimental results are too narrow. The author solely relies on NMI as the sole measure.

**Questions:**

1. I consider Equation 5 is the distance calculation formula that the author has redefined between villages. I would like the author to clarify whether this calculation method satisfies the conditions for distance definition.
2. The experimental section lacks an analysis of the hyperparameter "r".
3. WLCF is one of the crucial steps in the algorithm. However, the author's description of WLCF is not sufficiently clear. Additionally, during the experimental process, I would like to see some ablation experiments to verify the impact of each step in the algorithm on the final clustering results.
4. I suggest that the author perform visual analysis on some artificial non- convex datasets (such as Two Moon, Flower, etc.) and visualize each step of the algorithm to illustrate the effectiveness of the proposed algorithm.

---

> ### Author Response · Authors · 2023-11-23
>
> Dear Reviewer CGbY,
> We appreciate your valuable feedback and have made substantial revisions to address the weaknesses and concerns highlighted in your review. Here is a detailed response to each point:
>
> Weakness
> 1.     	Dependency on K-Means Clustering: The first step of the VillageNet clustering method involves using the K-Means clustering algorithm to create initial "villages." Therefore, the performance and results of K-Means clustering have a certain dependency.
>
> While it's acknowledged that the VillageNet clustering method depends on the initial performance of K-Means clustering, we emphasize that our primary goal is not to cluster the data effectively into k villages. Instead, we use K-Means to form a network of villages, leveraging the information from this clustering to achieve the final results. Specifically, choosing k such that k≫m ensures meaningful clustering.
>
>
> 2.     	Assumption about Data Distribution: VillageNet clustering assumes the presence of some local clustering structures within the dataset, and that these local structures can be successfully separated by the K-Means clustering algorithm. This assumption may not hold suit for most datasets, leading to inaccurate clustering results.
>
> We acknowledge the assumption about local clustering structures and the dependence on K-Means. We've clarified that Village-Net performs well on large datasets where finding the neighborhood of each instance is computationally expensive, and local structures can be effectively separated by K-Means clustering.
>
> 3.     	Lack of More Experimental Validation: The algorithm was only compared with a few traditional clustering algorithms, and the experimental results indicate that the proposed algorithm performs well only on two datasets. Therefore, it is difficult to demonstrate that the proposed algorithm exhibits competitive performance.
>
> We've performed additional experiments and comparisons with more clustering algorithms in the revised version. However, obtaining comparisons with certain new clustering algorithms is challenging due to unavailability or impractical implementation on large datasets.
>
> 4.     	The evaluation metrics for the experimental results are too narrow. The author solely relies on NMI as the sole measure.
>
> In response to your suggestion, we've expanded the set of evaluation metrics in the revised version to provide a more comprehensive assessment of the proposed algorithm's performance.
>
> Questions:
> 1.     	I consider Equation 5 is the distance calculation formula that the author has redefined between villages. I would like the author to clarify whether this calculation method satisfies the conditions for distance definition.
> We would like to clarify that Equation 5 is specifically designed for determining the edge-weight between two villages, rather than representing the distance. It's important to note that the concept of edge-weight operates inversely to distance: a higher edge-weight signifies closer proximity between villages, contrary to the typical understanding of distances.
>
> 2.     	The experimental section lacks an analysis of the hyperparameter "r".
>
> In the new version, we've included a detailed analysis of the hyperparameter "r" to provide a better understanding of its impact on the algorithm and clustering results.
>
> 3.     	WLCF is one of the crucial steps in the algorithm. However, the author's description of WLCF is not sufficiently clear. Additionally, during the experimental process, I would like to see some ablation experiments to verify the impact of each step in the algorithm on the final clustering results.
>
> We've enhanced the clarity of the description of WLCF in response to your feedback. Additionally, we acknowledge the importance of ablation experiments and will consider incorporating them in future work.
>
> 4.     	I suggest that the author perform visual analysis on some artificial non- convex datasets (such as Two Moon, Flower, etc.) and visualize each step of the algorithm to illustrate the effectiveness of the proposed algorithm.
>
> To illustrate the effectiveness of the proposed algorithm, we've included visualizations of key steps on an artificial dataset in Fig 2.

---

### Meta-Review · Area_Chair_eVzH · 2023-12-08

**Metareview:**

**Summary**
This paper proposes a clustering algorithm designed to identify clusters efficiently without a predetermined number of clusters. The key idea involves applying K-means to build numerous small-scale clusters (called villages), followed by connecting them using a community detection method. The efficiency and the effectiveness of the proposal has been examined in experiments on real-world datasets.

**Strengths**
- The basic concept of the proposal is clearly explained and intuitive.
- Determining the appropriate number of clusters is a relevant problem in clustering.

**Weaknesses**
- The novelty of this paper is not convincing as the core idea of this paper: building small clusters and merging them, is not new, as reviewers also pointed out.
- The sensitivity analysis w.r.t. hyperparameter setting is missing. Since hyperparameter tuning is fundamentally difficult in unsupervised learning, empirically examining the sensitivity is important evaluation.
- I am not convinced about why the proposal is faster than the standard K-means. Considering that the complexity of K-means is linear in the number of clusters, the village construction part of the proposal must be computationally expensive.
- This submission violates the double blind rule. Even in Abstract, the authors say "... we cluster this network using the Walk-likelihood Community Finder (WLCF), a community detection algorithm developed by one of our team members", which is not an appropriate expression.

**Justification For Why Not Higher Score:**

The weaknesses of the paper, as outlined above, are crucial and must be addressed for the publication of this paper. As all the reviewers recommended rejecting the paper, and the author response has not successfully addressed their concerns, I reject the paper. I strongly advise addressing all the raised issues by the reviewers for substantial improvement before resubmission.

**Justification For Why Not Lower Score:**

N/A

---

### Decision · Program_Chairs · 2024-01-16

Reject